# Design and Bioanalysis of Nanoliposome Loaded with Premium Red Palm Oil for Improved Nutritional Delivery and Stability

**DOI:** 10.3390/foods14040566

**Published:** 2025-02-08

**Authors:** Tanatchapond Rodsamai, Manat Chaijan, Prawit Rodjan, Arlee Tamman, Nassareen Supaweera, Mingyu Yin, Siriporn Riebroy Kim, Worawan Panpipat

**Affiliations:** 1Food Technology and Innovation Research Center of Excellence, School of Agricultural Technology and Food Industry, Walailak University, Nakhon Si Thammarat 80160, Thailand; Tanutchapornn24@gmail.com (T.R.); cmanat@wu.ac.th (M.C.); prawit.ro@wu.ac.th (P.R.); 2Thailand Institute of Nuclear Technology (Public Organization), Saimoon, Ongkarak District, Nakhon Nayok 26120, Thailand; arleet@tint.or.th; 3Health Sciences (International Program), College of Graduate Studies, Walailak University, Nakhon Si Thammarat 80161, Thailand; nassareen.sp@mail.wu.ac.th; 4College of Food Science and Technology, Shanghai Ocean University, No. 999, Huchenghuan Rd., Pudong New District, Shanghai 201306, China; myyin@shou.edu.cn; 5Food and Nutrition Program, Faculty of Agriculture, Kasetsart University, Bangkok 10900, Thailand; siriporn.r@ku.th

**Keywords:** anti-inflammatory activity, antioxidant, biological activity, encapsulation

## Abstract

Red palm oil (RPO), which is rich in carotenoids and tocotrienols, offers significant health-promoting properties. However, its utilization in functional foods is hindered by poor water solubility and instability under certain processing conditions. This study aimed to overcome these limitations by enhancing the bioactivity and stability of RPO through the ultrasound-assisted fabrication of nanoliposomes, formulated with varying ratios of egg yolk phosphatidylcholine (EYPC) to RPO. At a 3:1 ratio, the encapsulation efficiency (EE) began to reach >90%. Nanoliposome with the highest *β*-carotene EE (94.9%) (*p* < 0.05) and a typical oil loading content of 13.40% was produced by EYPC-to-RPO at a 7:1 ratio. As EYPC levels increased, the average vesicle size and the polydispersity index decreased, but the zeta potential and pH gradually increased. Nanoliposome prepared with an EYPC: RPO ratio of 3:1 showed the lowest peroxide value (PV) of 4.99 meqO_2_/kg, a thiobarbuturic acid reactive substances (TBARS) value of 0.20 mmol/kg, and greater 1,1-diphenyl-2-picrylhydrazyl radical (DPPH^•^) inhibition over 30 days of storage at 25 °C. All nanoliposomes showed anti-inflammatory activity without cell toxicity. Nanoliposomes present a promising delivery system for enhancing the biological activity and storage stability of RPO.

## 1. Introduction

Red palm oil (RPO) can be made by partially refining curd palm oil (CPO), resulting in a high concentration of biologically active vitamin A (carotene) and vitamin E (tocopherols and tocotrienols) [1]. RPO shares similarities with refined, bleached, and deodorized oils but retains its high content of carotenoids, giving it a distinctive red color. RPO contains 500–750 ppm of total carotenes, 600–1000 ppm of tocopherols and tocotrienols, 109–365 ppm of phytosterols, and minor components such as ubiquinone (18–25 ppm) and squalene (14–15 ppm) [2,3]. Its balanced fatty acid composition comprises 50% saturated fatty acids (e.g., palmitic acid and stearic acid), 40% monounsaturated fatty acids, and 10% polyunsaturated fatty acids (e.g., linoleic acid and linolenic acid). This composition allows RPO to remain in a semi-solid state and enhances its resistance to lipid oxidation compared to vegetable oils high in monounsaturated fatty acids [2]. RPO can be utilized as a dietary supplement or incorporated into various food products, particularly as a functional ingredient in lipid-based formulations such as cooking oil, margarine, spreads, gravy oil, cereal bars, snacks, and ice cream [2]. Its health benefits have been well-documented, including relieving vitamin A deficiency, increasing serum retinol levels, lowering cholesterol, boosting antioxidant levels, and reducing the risk of cancer [2,3].

RPO is commercially produced using molecular distillation for pre-treatment, acidification, and deodorization [2,4]. Nonetheless, carotenoids, tocopherols, and other minor active compounds might be partially destructive in primary refinement, necessitating efficient processes for retaining or concentrating these bioactive compounds. Alternative processes are still required to effectively retain micronutrients in RPO. One method for producing a premium RPO is microwave-assisted extraction [4]. Microwave energy provides the advantage of distributing heat throughout the oil palm fruit, enabling faster and more even deactivation of the lipase enzyme [5,6,7]. This results in greatly reduced free fatty acid (FFA) levels while maintaining a higher level of carotenoids and tocopherols than typical RPO, making it appropriate for food and pharmaceutical applications [4]. Vitamin A deficiency remains one of the most common dietary problems worldwide, especially in underdeveloped countries, where prevalence is higher than the global average. Premium RPO could be a good source of vitamin A deficiency prevention [8]. However, the active components in premium RPO, particularly carotenes, are unstable and prone to oxidation when exposed to light or high temperatures, resulting in nutritional impairment and rancidity. Furthermore, incorporating lipophilic RPO into aqueous food systems is difficult due to their limited solubility, low bioavailability, and sensitivity to light, heat, and oxygen. As a result, research into suitable encapsulation systems for premium RPO to improve bioavailability and stability is gaining momentum.

Liposomes, due to their lipidic property, can overcome the challenges of dispersing hydrophobic bioactive substances in water-based food formulations and their low bioavailability in the gastrointestinal tract, particularly in the small intestine [9]. Liposomes are spherical particles with diameters ranging from nanometers to micrometers, which form bilayer membranes with amphiphilic molecules, usually phospholipids [10]. They can encapsulate both hydrophilic and lipophilic molecules within their inner aqueous phase and lipid bilayer [11]. Liposomes are widely used as secure delivery mechanisms to enhance bioavailability, particularly for encapsulating bioactive food ingredients like carotenoids, unsaturated fatty acids, and phenolic compounds [12]. These vesicles offer several advantages: they are simple to prepare, possess a high level of biosafety, and can encapsulate both lipophilic and hydrophilic molecules. Furthermore, liposomes enable prolonged and regulated release times for compounds with low aqueous solubility and delayed digestion [13]. They also protect sensitive bioactive molecules from light and oxidative degradation [14]. Common phospholipids used in liposome formation include phosphatidylcholine, phosphatidylserine, phosphatidylethanolamine, phosphatidylglycerol, and phosphatidic acid, along with cholesterol, a sterol-type lipid. Due to their structural similarity to natural cell membranes, liposomes serve as valuable models for studying analyte–lipid membrane interactions. Their structural flexibility allows for the formation of a variety of crystalline structures [15,16]. Several studies have found that phospholipid content, as a wall material, has altered the functionality and stability of liposomes. As a result, a search for an appropriate liposome formulation to effectively encapsulate premium RPO is required.

In this study, premium RPO-loaded nanoliposomes were produced using different ratios of egg yolk phosphatidylcholine (EYPC) to RPO, using thin-film evaporation and sonication. RPO-loaded nanoliposomes were characterized in terms of size distribution, polydispersity index (PDI), zeta potential, encapsulation efficiency (EE), and loading content (LC). Furthermore, transmission electron microscopy (TEM) was used to examine the nanoliposome morphology, and Fourier-transform infrared (FTIR) spectroscopy was used to monitor changes in phospholipid arrangement following the incorporation of premium RPO. Storage stability was evaluated based on particle size, *β*-carotene retention rate, lipid oxidation, and antioxidant activity (DPPH assay). Finally, the cytotoxicity and nitric oxide (NO) inhibitory effects of premium RPO-loaded nanoliposomes were studied. This study would be useful in the development of bioavailable and stable RPO-loaded nanoliposomes capable of stabilizing *β*-carotene for potential applications as functional food ingredients.

## 2. Materials and Methods

### 2.1. Materials and Chemicals

Sangarun Palm Oil Co., Ltd. (Krabi, Thailand) provided premium red palm oil (RPO) extracted using a microwave-assisted method with 714.18 mg/kg of *β*-carotene, 584.92 mg/kg of α-tocopherol, 285.71 mg/kg of *β*-tocopherol, and 343.41 mg/kg of δ-tocopherol. DPPH (1,1-Diphenyl-2-picrylhydrazyl), EYPC, and other chemicals and reagents were purchased from Sigma-Aldrich (St. Louis, MO, USA). The cell line was grown in RPMI-1640 medium containing 2 mM glutamine (Gibco BRL, Gaithersburg, MD, USA), 10% *v*/*v* fetal bovine serum, and penicillin/streptomycin. The cells were kept at 37 °C with 5% CO_2_ humidity.

### 2.2. Fabrication of RPO-Loaded Nanoliposomes

RPO-loaded nanoliposomes were prepared using the thin-film evaporation and ultrasound-assisted method described by Song et al. [11]. EYPC was combined with RPO in various ratios, notably 1:1, 3:1, 5:1, 7:1, and 10:1 (*w*/*w*). Tween-80 and cholesterol were also added to the mixture in a set proportion of 1:1:1 (*w*/*w*/*w*) by weight of RPO. The empty nanoliposome without RPO was created as a control by combining EYPC, Tween-80, and cholesterol in a 1:1:1 ratio (*w*/*w*/*w*). The mixture was then dissolved in 30 mL of ethanol. After dissolution, the ethanol was evaporated under reduced pressure at 45 °C (BUCHI Rotavapor R-114, Flawil, Switzerland), resulting in a thin lipid film. To ensure complete solvent removal, the film was blown with nitrogen gas and hydrated for 10 min at 45 °C with 25 mL of distilled water. The liposomal suspension was then probe sonicated (Sonics & Materials Inc., Newtown, CT, USA) in an ice bath for 10 min at 450 W and 25% amplitude, with 5 s pulse-on and pulse-off intervals. The nanoliposomes were subsequently characterized and bioanalyzed.

### 2.3. Nanoliposome Characterization

#### 2.3.1. pH

The pH was measured using a Eutech pH meter (Singapore) calibrated with buffer solutions (pH 4.0 and 7.0). The results were based on three different nanoliposome measurements [17].

#### 2.3.2. Particle Size Distribution, Polydispersity Index (PDI), and Zeta Potential

The Zetasizer Nano ZS90 instrument (Malvern Instruments, Worcestershire, UK) was used to determine the particle size distribution, PDI, and zeta potential of naoliposomes via dynamic light scattering at a scattering angle of 90°. The nanoliposomes were 100-fold diluted in deionized water before the test, and each sample was equilibrated in the instrument for 2 min [18].

#### 2.3.3. EE and LC of *β*-Carotene

The EE was determined using the previous method [19], with minor modifications. The amounts of free and encapsulated RPO, expressed as *β*-carotene, were determined through extraction with an organic solvent and quantified using a UV-Vis spectrophotometer. Briefly, a moderate amount of sample was combined with 4 mL of hexane and mixed for 1 min before centrifugation at 5000 rpm for 3 min. The supernatant was collected after being extracted twice more with hexane, and the absorbance at 450 nm was measured against hexane as a blank. A standard curve was used to calculate the free *β*-carotene content (C_free_). The remaining liposomal suspension after extraction was mixed with 3 mL of ethanol. After demulsifying the solution for 10 min at 25 °C with the ultrasonic bath (40 kHz; Ultrasons-H model3000841JP Selecta, Barcelona, Spain), 4 mL of hexane was added. To obtain the loaded *β*-carotene in the nanoliposome (C_loaded_), the extraction and analysis were performed as described above for free *β*-carotene. The LC of *β*-carotene, defined as the amount of *β*-carotene incorporated into the liposome relative to the weight of EYPC used in the formulation, was determined as follows: free *β*-carotene was extracted using ethyl acetate and subjected to centrifugation at 5000 rpm for 10 min at 4 °C. This process was repeated three times. The resulting supernatants were pooled, and the *β*-carotene content was quantified. Equation (1) was used to calculate the EE of *β*-carotene, while Equation (2) was used to compute the loading content (LC, % *w*/*w*).(1)$${\mathrm{EE}}_{\beta C}\left(\%\right)=\frac{C_{\mathrm{loaded}}}{C_{\mathrm{loaded}}+C_{\mathrm{free}}}\times \;100$$(2)$$LC(\%w/w)=\frac{\mathrm{Incorporated}\;\mathrm{amount}\;\mathrm{of}\;\beta -\mathrm{carotene}}{\mathrm{Amount}\;\mathrm{of}\;\mathrm{EYPC}}\;\times \;100$$

#### 2.3.4. FTIR Spectra

The samples were lyophilized for at least 48 h before measurements using an LGJ-25C freeze drier (Sihuan Co., Ltd., Beijing, China). The samples were measured at room temperature using an FTIR-attenuated total reflection (ATR) spectrometer (Alpha, Bruker Co., Ettlingen, Germany) in the 400–4000 cm^−1^ range [20]. The lyophilized sample (5 mg) was placed onto the surface of the ATR crystal. The ATR crystal was thoroughly cleaned with ethanol and dried under nitrogen gas flow prior to analysis. The crystal was examined to ensure it was free of residue. Air was used to measure the background. The samples were scanned 32 times.

#### 2.3.5. TEM

The nanoliposome morphology was determined using a TEM (JEM-1400 Plus, Japanese Electronics Co., Ltd., Tokyo, Japan) and a negative staining method [21]. The sample was allowed to dry for 5 min on a copper grid coated in carbon. Filter paper was used to remove the excess material, and the grid was stained for an additional 3 min with a 1% sodium phosphotungstate solution. The excess liquid was then drained using filter paper.

### 2.4. Bioanalysis

#### 2.4.1. Cytotoxicity Assay

RAW264.7 cells are a well-known murine macrophage cell line that is frequently used to assess the possible toxicity of different substances on immune cells. They are an appropriate model for examining the effects of liposome formulations on cellular viability and function because of their capacity to replicate essential macrophage functions, including phagocytosis and cytokine generation [22,23]. The viability of RAW264.7 cells treated with the nanoliposomes was determined using the 3-(4,5-dimethylthiazol-2-yl)-2,5-diphenyltetrazolium bromide (MTT) test [22]. Prior to treatment, the cells were grown for 48 h until 70% confluence was achieved. The nanoliposomes were serially diluted to concentrations ranging from 0.98 to 250 g/mL before being incubated with the cells for 24 h. Each well was treated with 200 µL of 0.5 mg/mL MTT solution (Invitrogen, Life Technologies, Carlsbad, CA, USA) for the first 24 h of incubation, followed by a 4-h interval. The formazan crystals were dissolved in dimethyl sulfoxide. The absorbance (A) was measured using a microplate reader at 560 nm, and the background was removed at 670 nm. The cell viability was determined as follows:(3)$$Cell\;viability\left(\%\right)=[\frac{A_{\mathrm{treated}}}{A_{\mathrm{untreated}}}]\times 100$$

#### 2.4.2. Nitric Oxide (NO) Inhibitory Activity

NO inhibitory activity was used to evaluate the anti-inflammatory properties of the RPO-nanoliposomes. The Griess reagent was used to discover a stable NO end product called nitrite. Before being stimulated with 50 ng/mL LPS for 24 h, cells were preincubated for 1 h with dexamethasone (Dex), a commonly used drug, and various EYPC-to-RPO dosage ratios. A 96-well plate with 75 µL of culture supernatants, 65 µL of distilled water, and 10 µL of Griess reagent (1% sulfanilamide and 0.1% naphthylethylene in a 2.5% phosphoric acid solution) was then used. The absorbance at 540 nm was measured with a microplate reader after 30 min of incubation at 25 °C [23].

### 2.5. Storage Stability

The samples (~10 mL) were stored in sealed amber tubes at 25 °C in the dark for 30 days. The *β*-carotene retention rate, radical scavenging activity, lipid oxidation, and particle size distribution (as described in Section 2.3.2) were measured at predetermined intervals (0, 5, 10, 15, 20, 25, and 30 days) during storage.

#### 2.5.1. *β*-Carotene Retention Rate

The *β*-carotene content was analyzed using a spectrophotometer, as described in Section 2.3.3. The retention rate (%) of RPO-loaded nanoliposomes was calculated as follows:(4)$$\beta -carotene\;retention\;rate\left(\%\right)=\frac{\mathrm{Encapsulation}\;\mathrm{amount}\;\mathrm{of}\;\beta -\mathrm{carotene}\;\mathrm{after}\;\mathrm{storage}}{\mathrm{Encapsulation}\;\mathrm{amount}\;\mathrm{of}\;\mathrm{RPO}\;\mathrm{initially}\;\mathrm{prepared}}\;\times \;100$$

#### 2.5.2. Free Radical Scavenging Activity

The DPPH radical scavenging activity was measured according to Mohammadzadeh et al. [24]. The sample (80 μL) was mixed with 1 mL of a 90 µM methanolic DPPH solution and adjusted to a final volume of 4 mL by adding 95% methanol. The mixture was incubated in the dark at room temperature for 30 min. After incubation, the absorbance was measured at 517 nm. The DPPH radical inhibition (%) was calculated using the following equation:(5)$$DPPH\;radical\;inhibition\left(\%\right)=\left[\frac{A_{Blank}-A_{Sample}}{A_{Blank}}\right]\times 100$$

#### 2.5.3. Lipid Oxidation

The peroxide value (PV) and thiobarbituric acid reactive substances (TBARS) were used to determine the lipid oxidation of samples during storage. PV was reported as milliequivalents of free iodine per kg of lipid [12]. The TBARS was performed according to Song et al. [11]. In brief, 2.5 mL of the thiobarbituric acid (TBA) reagent (15% *w*/*v* trichloroacetic acid and 0.375% *w*/*v* TBA in 0.25 M HCl) was added to 1.0 mL of the nanoliposomes. After vortexing, the mixture was boiled for 15 min. The material was centrifuged for 10 min at 3000× *g* after cooling in an ice bath. The absorbance was determined using a spectrophotometer at 532 nm. The 1,1,3,3-tetraethoxypropane standard curve was established.

### 2.6. Statistical Analysis

A completely randomized design (CRD) was used in this investigation. The experiment was carried out in triplicate. The data were processed and plotted using GraphPad Prism 8.0 (GraphPad Software Inc., San Diego, CA, USA, 2018) and were represented as the means ± SD. All data were analyzed through one-way ANOVA using SPSS 16.0 for Windows (SPSS Inc., Chicago, IL, USA). Comparison of means was performed using Tukey’s multiple-range test.

## 3. Results and Discussion

### 3.1. Physicochemical Characteristics of RPO-Loaded Nanoliposomes

#### 3.1.1. Particle Size, PDI, Zeta Potential, and pH

The median particle size diameters of RPO-loaded nanoliposomes made with varying EYPC concentrations are shown in Table 1. As the ratio of EYPC to RPO increased (1:1–10:1), the average particle size of the nanoliposomes significantly decreased (*p* < 0.05). Adjusting the EYPC-to-RPO ratio from 1:1 to 10:1 resulted in nanoliposome particle sizes ranging from 179.3 ± 4.18 to 239.3 ± 7.22 nm (Table 1), indicating a significant impact on nanoliposome size. The particle sizes of empty nanoliposomes prepared without RPO were smaller than those of nanoliposomes loaded with RPO. However, all RPO-loaded nanoliposomes had particle sizes smaller than 240 nm, which could be referred to as nanoliposomes. Cui et al. [25] noticed that particles (<200 nm) had adequate bioavailability and consistent performance. Among the RPO-loaded nano liposomes, the smallest average particle size of 199 nm was observed in nanoliposomes made with a 10:1 EYPC-to-RPO ratio. It should be noted that lower RPO and higher EYPC contents resulted in a smaller median particle size diameter for the nanoliposome (Table 1). One possible explanation for this phenomenon is that the higher phospholipid content facilitates the formation of a sufficient and well-ordered lipid bilayer structure around a small number of RPO droplets. This was consistent with the findings of Song et al. [11], who noticed a decrease in liposome size as the content of phospholipid-loaded flaxseed oil increased. The PDI describes the uniformity of particle size distribution, with a lower PDI indicating a more uniform distribution [26]. All the samples had low PDI values, ranging from 0.345 to 0.396 (Table 1). Stable dispersions had PDI values less than 0.40 [11,12], implying that all nanoliposomes were relatively stable.

The stability of dispersion is determined by the zeta potential; a higher zeta potential value causes strong electrostatic repulsion between the particles, preventing the solution from flocculation and coalescence [11]. Stable particles possess a zeta potential greater than +30 mV or less than −30 mV [27]. All nanoliposomes had a zeta potential lower than −30 mV, ranging from −39.57 to −34.50 mV as pH increased from 4.60 to 4.72 (Table 1). Charged particles are electrostatically deposited on the surface of liposomes, resulting in changes in their charge [28]. The negative charge on all nanoliposomes indicated the realignment of anionic molecules at the particles’ surfaces. Liposomes with highly negative charge values are typically produced with phosphatidylcholine, a zwitterionic, amphiphilic molecule with a polar group that interacts with water molecules, resulting in a negative net surface charge [29]. This negative charge also contributes to stability in liposome dispersions, but it was insufficient to achieve complete stability. Ciont et al. [30] noticed a negative charge in nano-emulsion due to strong Van der Waals electrostatic repulsion, which maintained W/O system stability. A high surface charge promotes particle–cell contact, which improves bioactive chemical delivery. Surprisingly, increased EYPC content reduced the absolute zeta potential of nanoliposomes (Table 1). Since the pH of the solution ranged from 4.60 to 4.72, the phospholipid layer surface is made up of molecules with -PO^−^, -NH_3_^+^, and -COO^−^ groups in the aqueous solution side, which describe the equilibria between the bilayer and the solution ions and influence the net charge [31]. However, the zeta potentials of each experimental nanoliposome varied slightly. The highest absolute zeta potential was observed in RPO-loaded liposomes with EYPC-to-RPO ratios of 1:1 or 3:1 (*p* < 0.05). Overall, the nanoliposome system was relatively stable, as evidenced by a smaller median particle size, absolute zeta potential values greater than 30 mV, and dispersions with PDI less than 0.4.

#### 3.1.2. EE of *β*-Carotene and LC

EE is a significant aspect of nanocarrier performance, which improves the stability and preservation of drugs or other bioactive molecules [28]. The EE of *β*-carotene in nanoliposomes prepared with different EYPC-to-RPO ratios (1:1–10:1) was investigated. The EE of *β*-carotene and LC of RPO-loaded nanoliposomes with different ratios of EYPC to premium RPO was shown in Figure 1. The amount of EYPC employed had a substantial impact on the effectiveness of the RPO encapsulation. At a 3:1 EYPC-to-RPO ratio, EE started to go over 90%. The maximum EE of *β*-carotene (94.9%) was found for 7:1 EYPC to RPO, while the ratio 1:1 had the lowest EE (87.9%) (*p* < 0.05). A high EE indicates that an optimized proportion of the substance has been successfully incorporated into the nanoliposomes, which is desirable for efficient target delivery. However, a high EE may depend on various factors, including the properties of the substance being encapsulated, the composition of the nanoliposome formulation, the preparation method, and the conditions during encapsulation. In addition, Carneiro et al. [32] reported that a higher EE and more stable emulsions result in a decrease in the proportion of nonencapsulated material on particle surfaces. As the EYPC-to-premium RPO ratio increased (1:1 to 10:1), the LC of nanoliposomes considerably reduced (*p* < 0.05) except for the 7:1 ratio of EYPC to premium RPO. If excessive RPO is utilized, the membrane’s fluidity and instability may rise. Finally, the EE of *β*-carotene decreased as a result of the premium RPO leak [11]. However, these results showed that the encapsulation of bioactive substances in premium RPO did not affect the encapsulation effectiveness of any coating material in the nanoliposome.

The incorporation ability of *β*-carotene was primarily due to its location as a lipid-soluble component in a phospholipid bilayer hydrophobic core [33], which could easily reach saturation. Typically, *β*-carotene can align vertically with the membrane plane due to hydrogen bonding between their polar end groups and the membrane’s polar region. This vertical fashion contributed to a strong inserting ability by allowing more molecules to be inserted across the bilayer in addition to those present in the hydrophobic core. According to Van de Ven et al. [34], lutein had the best ability to insert into a liposomal membrane among carotenoids, followed by *β*-carotene, lycopene, and canthaxanthin.

#### 3.1.3. FTIR Spectra

FTIR is a widely used technique for analyzing the composition and structure of liposomes, providing insight into the functional groups found in the molecules. Figure 2 showed the FTIR spectra (4000–600 cm^−1^) of all RPO-loaded nanoliposomes and an empty nanoliposome. The peaks at 2924 cm^−1^ and 2854 cm^−1^ indicate symmetric stretching vibrations of the C-H group, C=O stretching of carboxylic acid groups at 1735 cm^−1^, N-H bending at 1541 cm^−1^, stretching bands of PO^2−^ in the range of 1250–1090 cm^−1^, and asymmetric stretching vibration of N(CH_3_)^3+^, causing the peak at 970 cm^−1^ [11]. There was a slight difference in peak height at the same wavenumber between RPO-loaded nanoliposomes and empty nanoliposomes, indicating that the liposome did not change significantly after encapsulation with different EYPC ratios. The comparable primary peaks of the RPO-loaded nanoliposomes and empty nanoliposomes illustrated that the encapsulation of premium RPO with a different EYPO content might not cause a chemical reaction or a change in RPO structure. Symmetric and asymmetric CH_2_ stretching vibrations in premium RPO-loaded nanoliposome (2854 cm^−1^ and 2923 cm^−1^) and empty nanoliposomes (2858 cm^−1^ and 2924 cm^−1^) were observed (Figure 2). It should be emphasized that premium RPO-loaded nanoliposomes altered the initial conformation of hydrocarbon chains and reduced the Van der Waals interactions between phospholipids [35]. The ester C=O groups are sensitive to polarity changes in their surroundings, resulting in the transition from 1735 cm^−1^ (empty nanoliposomes) to 1742 cm^−1^ (premium RPO-loaded nanoliposomes). The spectrum in this spectral area also exhibits strong stretches of ester functional groups from lipids and fatty acids, indicating the overall amount of lipids in the oil [36]. The presence of *β*-carotene in RPO-loaded nanoliposomes caused the symmetric and asymmetric stretching vibrations of PO^2−^ in empty nanoliposomes (1080 cm^−1^ and 1247 cm^−1^) to shift to higher wave numbers (1094 cm^−1^ and 1238 cm^−1^). Due to its hydrophobicity, *β*-carotene exhibits triplet peaks in the high-wave number region (3000 cm^−1^–2800 cm^−1^), with traces found at wavelengths of 1086 cm^−1^ and 1045 cm^−1^ in plane bending modes [37]. Furthermore, the empty nanoliposome and RPO-loaded nanoliposomes displayed distinct stretching bands of the C-H bond in CH_2_. This band appears at 882 cm^−1^ and 838 cm^−1^ in the spectra of empty and RPO-loaded nanoliposomes, respectively.

### 3.2. Morphological Structure

TEM is a powerful imaging technique for observing the internal structure and morphology of materials with nanometer resolution. Empty nanoliposomes (without premium RPO) had a non-spherical structure, increased irregularity in the vesicle membrane, large particle size with a heterogeneous size distribution, and a slightly rough surface (Figure 3). An increasing EYPC ratio reduced the particle size of the experimental nanoliposomes. The nanoliposomes were uniformly spherical or nearly round, with few obvious breaks or adhesions and evenly dispersed without visible aggregation or fusion. Furthermore, a low phospholipid level in the cell wall may not prevent lipid molecules within the nanoliposome. However, nanoliposomes can exhibit various shapes depending on their composition, preparation method, and external conditions.

### 3.3. Storage Stability

#### 3.3.1. Retention of *β*-Carotene

The retention of *β*-carotene is a crucial metric for assessing the protective role of nanoliposomes in encapsulating RPO, accurately reflecting the change in bioactive activity in nanoliposomes over time [26]. Figure 4a shows the retention of *β*-carotene in RPO-loaded nanoliposomes stored at 25 °C under the dark for 30 days. The retention rate of *β*-carotene in all samples decreased over time. The highest retention of *β*-carotene in nanoliposomes was observed at a 3:1 ratio of EYPC to RPO, followed by 5:1, 10:1, 7:1, and 1:1 ratio. This study found that the ratio of EYPC to RPO significantly impacts the *β*-carotene retention rate. It might be due to the function of wall material in shielding core compounds from environmental elements that cause deterioration, specifically temperature, light, and oxygen [38]. Proper EYPC content resulted in a strong wall surrounding the RPO counterpart, which protected *β*-carotene from light and oxygen [37]. However, the high EYPC content resulted in the formation of a looser wall around the RPO, possibly due to strong intermolecular interaction between EYPC molecules. The presented results suggested that using RPO-loaded nanoliposomes at the proper ratio could significantly improve the storage stability of *β*-carotene.

#### 3.3.2. Vesicle Sizes, PDI, and Zeta Potential

This depicts the changes in the average particle size of empty nanoliposomes (without premium RPO) and RPO-loaded nanoliposomes during 30 days of storage (Figure 4b). The average vesicle size increased from 239.27 nm to 339.89 nm in the premium RPO-loaded nanoliposome at a 1:1 ratio of EYPC to RPO, whereas other RPO-loaded nanoliposomes had a mean particle size of less than 300 nm throughout the 30-day storage period (Figure 4b). An increased EYPC content resulted in a smaller change in the medium particle size of RPO-loaded nanoliposomes, indicating more stable dispersion. This could be due to a thicker particle wall with more EYPC, which results in a stable particle. Chen et al. [39] reported that phospholipids in lipid bilayers naturally rearrange to achieve the most stable configuration. As a result, the nanoliposome particle size variation increased or decreased with storage time until the nanoliposomes reached a stable state, at which point the ultimate particle size of the nanoliposomes became the most stable.

The PDI can be used to describe the particle size distribution; the lower the PDI, the more uniform the particle size distribution [26]. It was reported that the PDI of the dispersions stayed below 0.4, demonstrating that the nanoliposome system was mostly stable [11]. The PDI of all RPO-loaded nanoliposomes was slightly changed but remained less than 0.4 (Figure 4c). A higher EYPC concentration resulted in a lower PDI, indicating a smaller change in particle size distribution. These results were consistent with the changes in the mean particle size of the tested nanoliposomes (Figure 4b). It should be noted that the change in PDI value was largely determined by the ratio of EYPC to premium RPO.

Figure 4d depicts the changes in the zeta potentials of empty nanoliposomes and premium RPO-loaded nanoliposomes prepared with different EYPC to PPO ratios during storage. For EYPC:RPO ratios of 1:1, 3:1, 5:1, 7:1, and 10:1, the zeta potentials of RPO-loaded nanoliposomes were −41.57, −41.33, −40.00, −40.57, −41.73, and −42.67 mV, respectively, over a 30-day period. The results confirmed that the nanoliposomes were stable, with a zeta potential greater than +30 mV or less than −30 mV [20]. This was likely due to the fact that a higher zeta potential can effectively prevent aggregation and precipitation by ensuring adequate repulsion between nanoliposomes. In summary, using an EYPC-to-RPO ratio of 3:1 or higher can produce highly stable RPO-loaded nanoliposomes.

#### 3.3.3. DPPH^•^ Scavenging Activity and Lipid Oxidation

This depicts the DPPH^•^ inhibitory activity of nanoliposomes containing varying ratios of EYPC to premium RPO over 30 days of storage (Figure 5a). The DPPH^•^ scavenging activity of all nanoliposomes decreased with increased storage time (*p* < 0.05). Nanoliposomes with EYPC-to-premium RPO ratios of 1:1 showed the lowest DPPH^•^ scavenging activity (*p* < 0.05). Increased EYPC content resulted in increased radical scavenging activity, indicating that EYPC scavenged free radicals (Figure 5a). According to Zabodalova et al. [40], liposomal *β*-carotene contains two active components: *β*-carotene and EYPC as the wall material, which contribute to the additive’s hepato-stimulation activity. The other explanation is associated with the thicker EYPC wall formed by the higher EYPC content, which could potentially protect *β*-carotene. This resulted in the greater radical scavenging activity associated with the larger retention of *β*-carotene, as shown in Figure 4a. RPO-loaded nanoliposomes with a high amount of remaining *β*-carotene showed increased antioxidant activity, which has been linked to anticancer activity, heart disease risk reduction, and cataract prevention [19]. The optimal EYPC ratio for nanoliposome wall coating can delay *β*-carotene deterioration, leading to high radical scavenging activity.

The quality and stability of nanoliposomes may be significantly influenced by the lipid oxidation process. The changes in PV and TBARS over storage time of various RPO-loaded nanoliposomes are shown in Figure 5b,c, respectively. The PV (Figure 5b) and TBARS (Figure 5c) increased significantly as the storage time increased (*p* < 0.05). Nanoliposomes prepared with a 1:1 EYPC-to-RPO ratio had the highest PV and TBARS values upon 30 days of storage (*p* < 0.05), with maximum values of 6.05 meqO_2_/kg and 0.22 mmol/kg, respectively. This result was consistent with the remaining *β*-carotene (Figure 4a) and radical scavenging activity (Figure 5a) in this ratio. As expected, the empty nanoliposome exhibited the least lipid oxidation over the storage time (*p* < 0.05). The PV of this nanoliposome was within the acceptable range (10 meqO_2_/kg). Lipid oxidation may cause auto-oxidative degradation of *β*-carotene during the experiment. However, the result suggests a link between lipid oxidation and *β*-carotene depletion. Lipid oxidation is influenced by two major mechanisms: the direct reaction of *β*-carotene with pro-oxidant elements rather than lipids, and the effects of *β*-carotene on membrane chemical and physical properties oxidation [33]. In contrast, the strong lipid oxidation inhibition activity of *β*-carotene was primarily attributed to its effects on lipid physicochemical properties other than physical oxidation, owing to the likely stiffening effects on the liposomal membrane.

### 3.4. Biocompatibility and In Vitro Anti-Inflammatory Activity of Nanoliposomes

The biocompatibility of RPO-loaded nanoliposomes and empty nanoliposomes with a varying EYPC content was assessed against RAW264.7 murine macrophages cell (Figure 6). The empty nanoliposome (without RPO) had the greatest effect on cell viability when compared to the premium RPO-loaded nanoliposomes at doses ranging from 0.98 to 250 µg/mL. The RPO-loaded nanoliposomes with various EYPC tested contents (1:1 to 10:1, *w*/*w*) showed non-toxicity and relatively good biocompatibility. The reduction in cell viability at higher liposome concentrations may be attributed to several potential mechanisms, including phospholipid concentration, oxidative stress, physical crowding, and nutrient deprivation. Elevated concentrations of liposomes result in higher levels of phospholipids, which could disrupt cellular membranes and lead to cytotoxicity. This is consistent with previous studies reporting concentration-dependent cytotoxic effects of lipid-based carriers at higher doses. For instance, lipid nanocapsules have been shown to exhibit toxicity at high concentrations, with their effects varying in a cell-type-dependent manner [41]. Similar trends have been observed in liposomal delivery systems, such as anti-tubercular agent-loaded liposomal vesicles [42], where cytotoxic effects were attributed to the same factors.

Notably, the cytotoxicity threshold often varies depending on the cell type, liposome composition, and incubation conditions, highlighting the need for further investigation of our specific formulation [43]. At higher concentrations, the interaction of liposomes with cell membranes could induce oxidative stress, impairing cellular function and viability. This effect is likely influenced by the intricate mechanisms underlying the interaction between liposomes and cell membranes, which vary depending on the composition and characteristics of the liposomes [44]. While premium RPO-loaded nanoliposomes contain *β*-carotene with known antioxidant properties, the empty nanoliposomes lack such bioactives, potentially making them more prone to oxidative damage at elevated doses. In addition to these mechanisms, high liposome concentrations in the culture medium may alter the local environment, leading to physical crowding or reduced nutrient and oxygen availability for the cells, which could impair growth.

Although high concentrations may present challenges, it is important to note that the effective concentrations required for functional applications, such as delivering bioactive compounds, are often below the threshold that induces cytotoxicity. This underscores the potential of RPO-loaded nanoliposomes for safe use when appropriately dosed. The effect of nanoliposomal concentration on NO production in RAW 264.7 cells is depicted in Figure 7. All RPO-loaded nanoliposomes demonstrated anti-inflammatory activity by inhibiting NO production (Figure 7), whereas the empty nanoliposomes showed no such effect. Mokdad et al. [45] reported that phospholipids found in the liposomal membrane may have an added anti-inflammatory action, potentially enhancing their bioavailability. While certain studies have established that phospholipids possess anti-inflammatory properties, the results of this study indicated that empty nanoliposomes did not inhibit NO production. This suggests that their anti-inflammatory potential might be influenced by specific conditions or the presence of additional components.

The lack of observable anti-inflammatory activity in empty nanoliposomes can be attributed to the absence of bioactive compounds, which are likely essential for eliciting such effects. Moreover, the concentration of phospholipids in the formulation may have been insufficient to independently trigger an anti-inflammatory response. This finding is consistent with the literature, which highlights the dose-dependent nature of phospholipid activity in exhibiting anti-inflammatory properties [46].

According to Li et al. [47], *β*-carotene regulates tissue metabolism, which includes microbial flora, oxidative stress, and resistance to inflammation. It interacts with several inflammatory targets, making it a potential therapeutic agent for a variety of diseases. *β*-carotene can reduce inflammation by inhibiting the production of NO, prostaglandin E2, and superoxide dismutase [47]. It also reduces the expression of iNOS/cox-/NADPH oxidase proteins and mRNA, and inhibits TNF-α levels [48]. *β*-Carotene can boost the number of NK cells in Wistar rats’ blood, increase IL-2 and TNF-α levels, increase GSH-Px in the liver, inhibit tumor cell growth, and lower ALT and AST levels in mice with liver cancer [49]. Li et al. [47] noticed that *β*-carotene reduced LPS-induced oxidation in porcine intestinal epithelial cells, inhibited Caspase-3 expression, and reduced inflammatory responses in macrophages by inhibiting NF-kB, JK2/STAT3, and JNK2/p38MAPK signaling pathways.

## 4. Conclusions

The premium RPO was successfully encapsulated inside a nanoliposome made of egg phosphatidylcholine (EYPC), cholesterol, and Tween 80. The ratio of EYPC to premium RPO was critical for encapsulation. The optimal ratio for EYPC to premium RPO was 7:1, which resulted in the highest encapsulation efficiency and the smallest average particle size. The oxidation stability of RPO was significantly improved after nanoencapsulation with the appropriate EYPC ratio. The RPO-loaded nanoliposomes had a spherical shape with a uniform size distribution. RPO-loaded nanoliposomes with varying EYPC ratios demonstrated anti-inflammatory activity without cytotoxic effects against RAW264.7 cells. Furthermore, this study showed that RPO-loaded nanoliposomes had longer-term bioactivity than empty nanoliposomes. These findings highlight nanoliposomes as a promising delivery system for enhancing the biological activity and storage stability of premium RPO. The results of this study have practical applicability in the development of functional foods and nutraceuticals, particularly in improving the stability and bioavailability of lipid-soluble bioactives such as RPO. The successful encapsulation of RPO into nanoliposomes suggests potential use in industries where oxidative stability and sustained bioactivity are critical, such as in dietary supplements, food fortification, and pharmaceutical formulations.

Future research should address current limitations by further investigating the stability and bioactivity of RPO-loaded nanoliposomes under diverse environmental conditions, such as heat, light, and pH, as well as during long-term storage. Such studies would provide valuable insights into their robustness and practical applicability. Additionally, exploring complementary approaches, including the incorporation of other bioactive compounds or antioxidants into the nanoliposomes, could enhance their functionality and broaden their potential applications. Testing these nanoliposomes in broader contexts, such as in vivo models or within complex food matrices, will be essential to validate their efficacy, safety, and suitability for real-world use.

## Figures and Tables

**Figure 1 foods-14-00566-f001:**
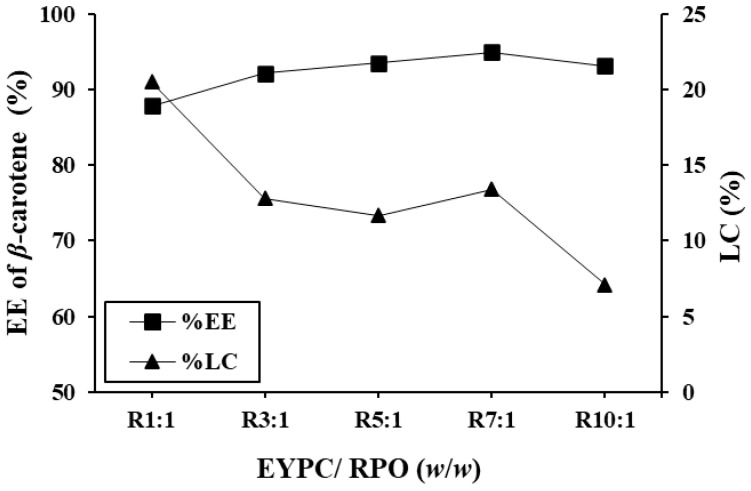
Encapsulation efficiency (EE) of *β*-carotene and loading content (LC) of premium-grade red palm oil (RPO)-loaded nanoliposomes at different ratios of egg yolk phosphatidylcholine (EYPC) to premium-grade RPO. The bars reflect the standard deviation of triplicate determinations.

**Figure 2 foods-14-00566-f002:**
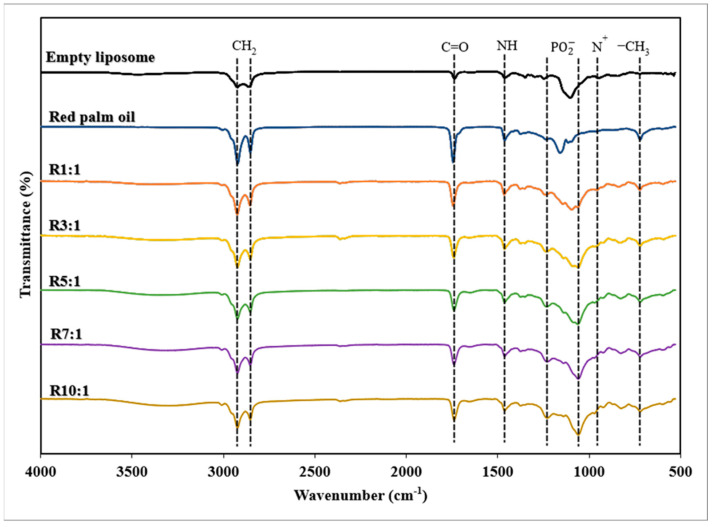
Fourier-transform infrared (FTIR) spectra of empty nanoliposome, premium-grade red palm oil (RPO), and RPO-loaded nanoliposomes at different ratios of egg yolk phosphatidylcholine (EYPC) to premium-grade RPO. The EYPC-to-RPO (*w*/*w*) ratio ranged from 1:1 to 10:1, as mentioned.

**Figure 3 foods-14-00566-f003:**
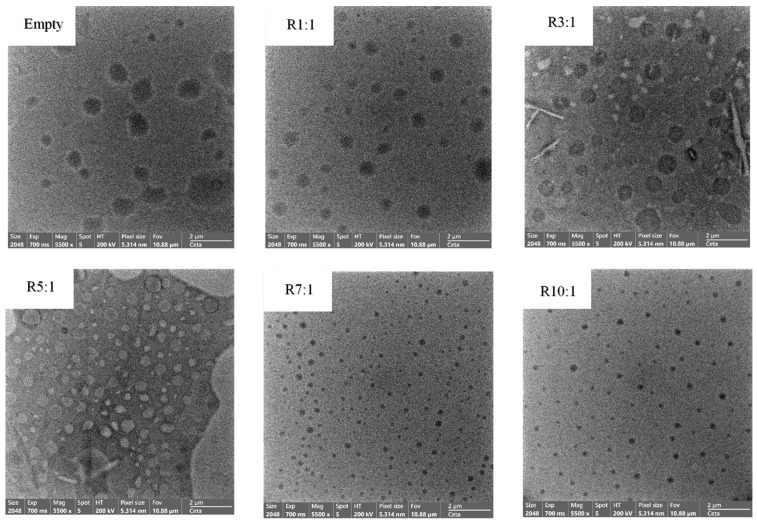
Transmission electron microscopy (TEM) images of empty liposome and premium-grade red palm oil (RPO)-loaded nanoliposomes at different ratios of egg yolk phosphatidylcholine (EYPC) to premium-grade RPO. The EYPC-to-RPO (*w*/*w*) ratio ranged from 1:1 to 10:1, as mentioned. The magnification was 5500×.

**Figure 4 foods-14-00566-f004:**
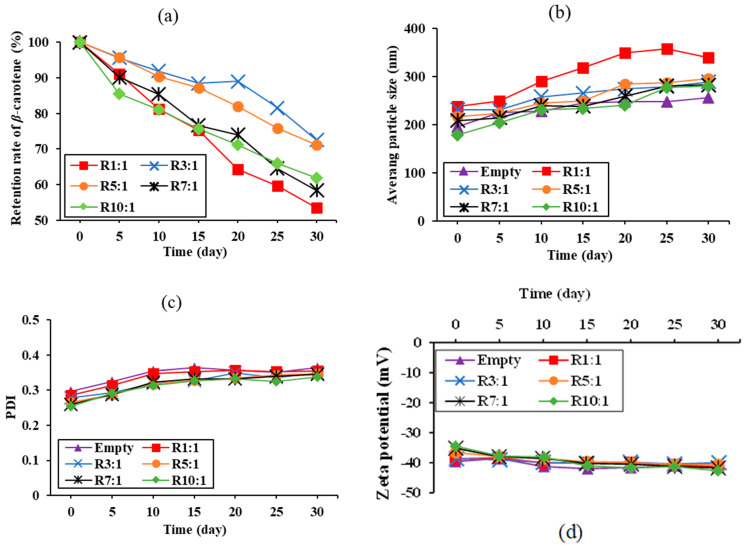
Variations of red palm oil retention rate (**a**), vesicle sizes (**b**), polydispersity index (PDI) (**c**) and zeta potential (**d**) of premium-grade red palm oil (RPO)-loaded liposomes at different ratios of egg yolk phosphatidylcholine (EYPC) to premium-grade RPO during 30 days’ storage at 25 °C.

**Figure 5 foods-14-00566-f005:**
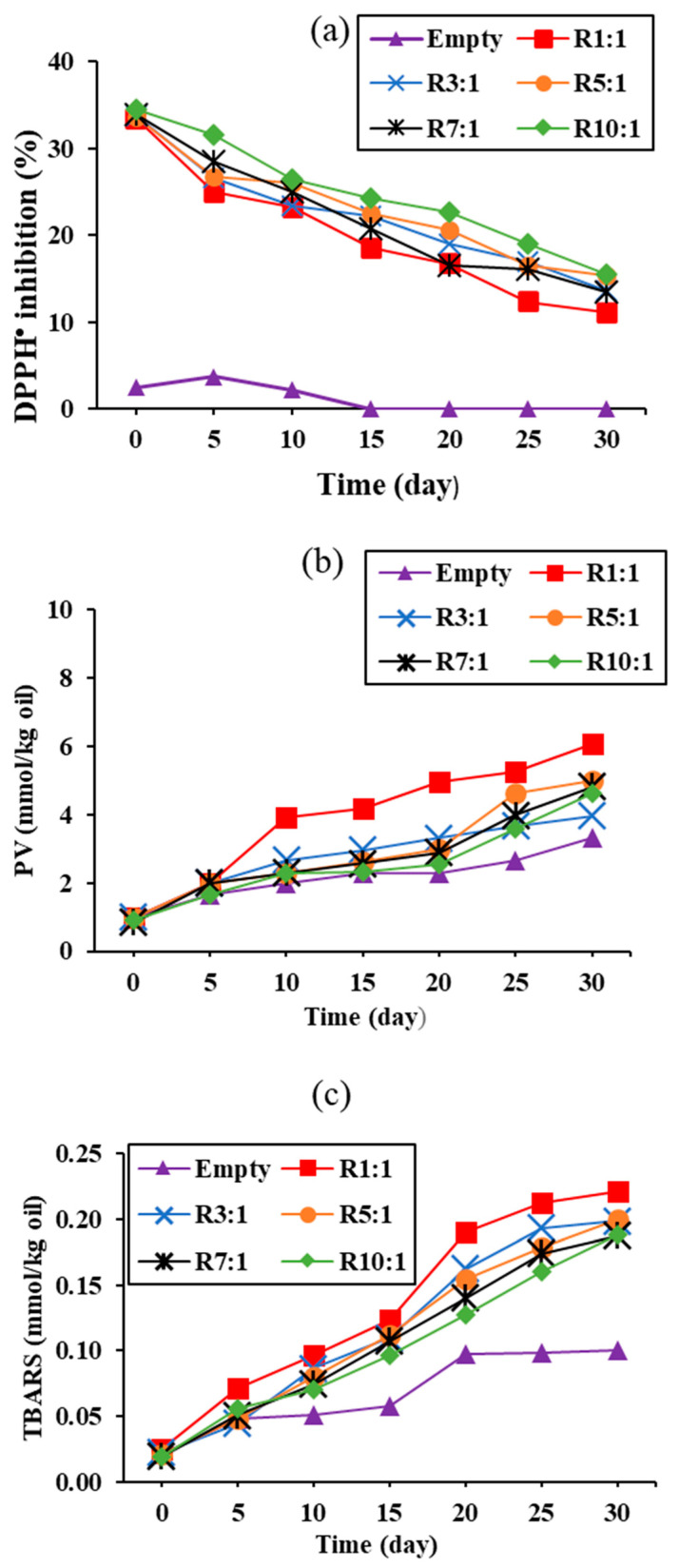
Variations of DPPH^•^ inhibition (**a**), peroxide value (PV) (**b**), and thiobarbituric acid reactive substances (TBARS) (**c**) of premium-grade red palm oil (RPO)-loaded liposomes at different ratios of egg yolk phosphatidylcholine (EYPC) to premium-grade RPO during 30 days of storage at 25 °C.

**Figure 6 foods-14-00566-f006:**
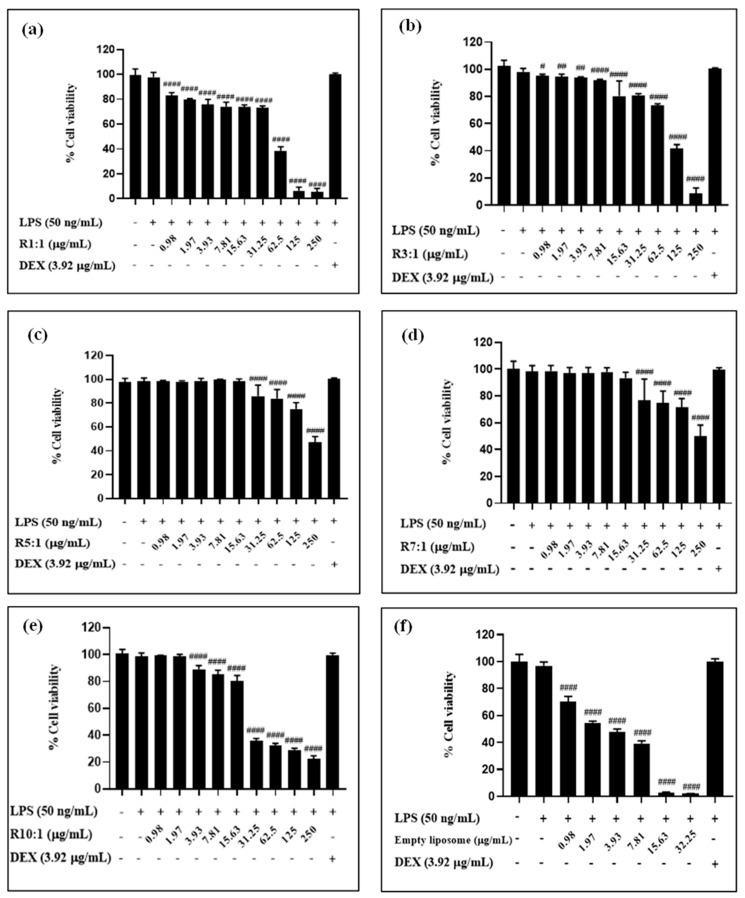
Cell viability of RAW264.7 macrophages treated with premium-grade red palm oil (RPO)-loaded liposomes at different ratios of egg yolk phosphatidylcholine (EYPC) to premium-grade RPO, namely R1:1 (**a**), R3:1 (**b**), R5:1 (**c**), R7:1 (**d**), R10:1 (**e**) and empty liposome (**f**). Each bar graph represents the means ± standard deviation. The #, ##, #### symbols indicate significant differences at *p* < 0.0332, 0.0021, and 0.0001 as compared to the untreated cells (control), determined using the MTT assay.

**Figure 7 foods-14-00566-f007:**
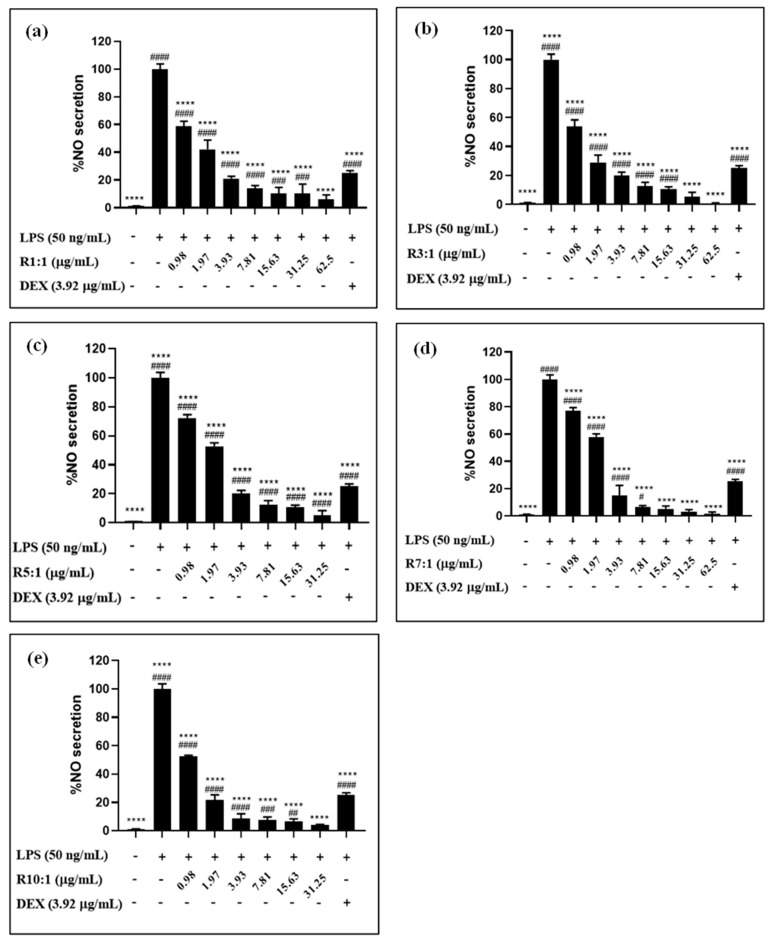
Nitric oxide (NO) inhibitory activity of premium-grade red palm oil (RPO)-loaded nanoliposomes at different ratios of egg yolk phosphatidylcholine (EYPC) to premium-grade RPO, namely R1:1 (**a**), R3:1 (**b**), R5:1 (**c**), R7:1 (**d**), and R10:1, and (**e**), in LPS-activated RAW264.7 macrophages with respect to their cell viability. Each bar graph represents the means ± standard deviation. The #, ##, ###, #### symbols indicate significant differences at *p* < 0.0332, 0.0021, 0.0002, and 0.0001 as compared to the untreated cells (control), whereas **** indicate the significant differences at *p* < 0.0001 as compared to the LPS-treated cells (LPS). The statistical analyses were conducted using one-way ANOVA with Dunnett’s multiple comparison test. Empty liposome had no NO inhibitory activity.

**Table 1 foods-14-00566-t001:** Average vesicle size, polydispersity index (PDI), zeta potential, and pH of premium-grade red palm oil (RPO)-loaded nanoliposomes at different ratios of phosphatidylcholine to premium-grade RPO.

EYPC/RPO (*w*/*w*)	Average Vesicle Size (nm)	PDI	Zeta Potential (mV)	pH
Empty nanoliposome	196.20 ± 2.16 a	0.396 ± 0.003 e	−39.57 ± 0.06 a	4.60 ± 0.006 a
1:1	239.3 ± 7.22 d	0.385 ± 0.004 d	−38.73 ± 0.65 b	4.65 ± 0.000 b
3:1	231.6 ± 5.65 d	0.378 ± 0.004 c	−38.63 ± 0.21 b	4.66 ± 0.000 c
5:1	217.2 ± 3.84 c	0.366 ± 0.002 b	−36.60 ± 0.10 c	4.68 ± 0.006 d
7:1	209.0 ± 2.60 c	0.359 ± 0.003 a	−35.17 ± 0.50 d	4.67 ± 0.006 c
10:1	199.3 ± 4.18 b	0.354 ± 0.006 a	−34.50 ± 0.56 d	4.72 ± 0.000 d

Values are presented as the mean ± standard deviation of three measurements. Letters within the same column are compared for significant differences (*p* < 0.05).

## Data Availability

The original contributions presented in the study are included in the article; further inquiries can be directed to the corresponding author.

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
