# Peer review of "Design and Bioanalysis of Nanoliposome Loaded with Premium Red Palm Oil for Improved Nutritional Delivery and Stability"

_foods, 2025, doi:10.3390/foods14040566_

Round 1
Reviewer 1 Report
Comments and Suggestions for Authors
Dear authors, this manuscript provides insightful findings on relevant and innovative topic regarding loaded nanoliposomes. The work is evident, and the discussion is particularly well explained. However, there is still some room for improvement, especially in the introduction section and some parts of the results. Comments are listed below:
· - The abstract is overly concise and would benefit from the addition of a few introductory sentences to establish context and emphasize the significance of the research, rather than merely presenting a summary of the results. I suggest broadening it to provide more than just an objective of this study.
· - A similar issue can be observed in the introduction section, which lacks a theoretical overview of red palm oil, including its value and bioactivity. Additionally, it is overly focused on microwave extraction, which is not the primary topic of this study. This emphasis appears unnecessary and could be replaced with more relevant information to improve the text's coherence and better explain the need for the research.
· - Additionally, the text from lines 46–54 requires revision, as its wording resembles a description of the materials and methods used for microwave extraction, rather than serving as part of the introduction related to red palm oil and the relevant literature findings associated with this study.
· - Furthermore, avoid excessive repetition, such as starting multiple sentences with "Liposomes are...," to maintain a more varied and engaging narrative flow.
· - Line 136- EE was expressed using measured free and encapsulated β-carotene content. Revise wording and fix α-carotene to β-carotene if that is an accidental mistake.
· - Regarding the measurement of LC, provide a detailed explanation of what is meant by the incorporated amount of carotenoids. Clarify whether the amount of lipid is directly measured or if the calculations are based solely on the formulation of the lipid particles during their preparation.
· - Section 2.5.2 should be expanded to ensure consistency with the rest of the manuscript. Provide, in short, additional details about the DPPH antioxidant test, to align with the other sections
· - Line 206- After vertexing –> vortexing
· - Please include a scale bar in the TEM figures to ensure clarity and accurate interpretation of the results. Additionally, specify the level of magnification below figure 3.
· - Figure 6 presents intriguing findings on the biocompatibility of RPO-loaded nanoliposomes and empty nanoliposomes. How do you account for the decrease in cell viability at higher liposome concentrations (125 and 250 µL/mL)? Would this reduction be considered a limitation of the formulation? To enhance this section, provide a more detailed discussion, including potential mechanisms behind the cytotoxicity at higher concentrations. Additionally, strengthen the comparison with existing literature by highlighting similar findings or contrasting trends and their implications for the application of these nanoliposomes.
· - In line 470 it is mentioned that empty nanoliposome displayed no 470 NO inhibition, but some researchers reported that phospholipids may have an added anti-inflammatory action. Did the authors consider their results to not be in accordance with this, or that without loaded bioactive compounds there is no added antiinflamatory activity. Try to incorporate your results with literature more cohesively. This also go for next paragraph, which seems to be a bit nepoveziv.
· - In line 470, it is stated that empty nanoliposomes exhibited no NO inhibition; however, studies that you have mentioned (41) said that phospholipids may have an added anti-inflammatory action. Do the authors interpret their findings as being inconsistent with these reports, or do they attribute the lack of anti-inflammatory activity to the absence of bioactive compounds in the empty nanoliposomes? Consider integrating your results more cohesively with the existing literature, addressing potential reasons for this discrepancy. This recommendation also applies to the following paragraph, which appears somewhat disconnected. Strengthen the narrative flow and ensure clear connections between the results and relevant literature.
· - In the conclusions, please include a discussion of the practical applicability of the obtained results, emphasizing their potential impact or real-world use. Additionally, outline directions for future research, such as addressing current limitations, exploring complementary approaches, or testing the findings in broader or more applied contexts.
Author Response
Reviewer 1
Dear authors, this manuscript provides insightful findings on relevant and innovative topic regarding loaded nanoliposomes. The work is evident, and the discussion is particularly well explained. However, there is still some room for improvement, especially in the introduction section and some parts of the results. Comments are listed below:
- - The abstract is overly concise and would benefit from the addition of a few introductory sentences to establish context and emphasize the significance of the research, rather than merely presenting a summary of the results. I suggest broadening it to provide more than just an objective of this study.
Ans: It was revised as requested. “Red palm oil (RPO), rich in carotenoids and tocotrienols, offers significant health-promoting properties. However, its utilization in functional foods is hindered by poor water solubility and instability under certain processing conditions. This study aimed to overcome these limitations by enhancing the bioactivity and stability of RPO through the ultrasound-assisted fabrication of nanoliposomes, formulated with varying ratios of egg yolk phosphatidylcholine (EYPC) to RPO.”
- - A similar issue can be observed in the introduction section, which lacks a theoretical overview of red palm oil, including its value and bioactivity. Additionally, it is overly focused on microwave extraction, which is not the primary topic of this study. This emphasis appears unnecessary and could be replaced with more relevant information to improve the text's coherence and better explain the need for the research.
Ans: An review of RPO, including its value and bioactivity, was added. “RPO shares similarities with refined, bleached, and deodorized oils but retains its high content of carotenoids, giving it a distinctive red color. RPO contains 500-750 ppm of total carotenes, 600-1000 ppm of tocopherols and tocotrienols, 109-365 ppm of phytosterols, and minor components such as ubiquinone (18-25 ppm) and squalene (14-15 ppm) [2, 3]. Its balanced fatty acid composition comprises 50% saturated fatty acids (e.g., palmitic acid and stearic acid), 40% monounsaturated fatty acids, and 10% polyunsaturated fatty acids (e.g., linoleic acid and linolenic acid). This composition allows RPO to remain in a semi-solid state and enhances its resistance to lipid oxidation compared to vegetable oils high in monounsaturated fatty acids [2]. RPO can be utilized as a dietary supplement or incorporated into various food products, particularly as a functional ingredient in lipid-based formulations such as cooking oil, margarine, spreads, gravy oil, cereal bars, snacks, and ice cream [2]. Its health benefits have been well-documented, including relieving vitamin A deficiency, increasing serum retinol levels, lowering cholesterol, boosting antioxidant levels, and reducing the risk of cancer [2, 3].”
Microwave extraction was introduced as a method of producing superior RPO with low free fatty acid level and high bioactive component content, according to reports. In this study, commercially available premium RPO produced using the microwave extraction technique was utilized. However, the explanation of microwave extraction was shortened as requested.
- - Additionally, the text from lines 46–54 requires revision, as its wording resembles a description of the materials and methods used for microwave extraction, rather than serving as part of the introduction related to red palm oil and the relevant literature findings associated with this study.
Ans: Done as requested. “Microwave energy provides the advantage of distributing heat throughout the oil palm fruit, enabling for faster and more even deactivation of the lipase enzyme [5, 6, 7]. This resulted in much reduced free fatty acid (FFA) levels while maintaining a higher level of carotenoids and tocopherols than typical RPO, making it appropriate for food and pharmaceutical applications [4].”
- - Furthermore, avoid excessive repetition, such as starting multiple sentences with "Liposomes are...," to maintain a more varied and engaging narrative flow.
Ans: Done as requested.
- - Line 136- EE was expressed using measured free and encapsulated β-carotene content. Revise wording and fix α-carotene to β-carotene if that is an accidental mistake.
Ans: Thank you very much. It was revised to “The amounts of free and encapsulated RPO, expressed as β-carotene, were determined through extraction with an organic solvent and quantified using a UV-Vis spectrophotometer.”
- - Regarding the measurement of LC, provide a detailed explanation of what is meant by the incorporated amount of carotenoids. Clarify whether the amount of lipid is directly measured or if the calculations are based solely on the formulation of the lipid particles during their preparation.
Ans: It was revised to “The LC of β-carotene, defined as the amount of β-carotene incorporated into the liposome relative to the weight of EYPC used in the formulation, was determined as follows: free β-carotene was extracted using ethyl acetate and subjected to centrifugation at 5000 rpm for 10 min at 4 °C. This process was repeated three times. The resulting supernatants were pooled, and the β-carotene content was quantified. Equation (1) was used to calculate the EE of β-carotene, while equation (2) was used to compute the loading content (LC, % w/w).”
- - Section 2.5.2 should be expanded to ensure consistency with the rest of the manuscript. Provide, in short, additional details about the DPPH antioxidant test, to align with the other sections
Ans: The detail for DPPH antioxidant test was given as requested.
- - Line 206- After vertexing –> vortexing
Ans: Done as requested.
- - Please include a scale bar in the TEM figures to ensure clarity and accurate interpretation of the results. Additionally, specify the level of magnification below figure 3.
Ans: Done as requested.
- - Figure 6 presents intriguing findings on the biocompatibility of RPO-loaded nanoliposomes and empty nanoliposomes. How do you account for the decrease in cell viability at higher liposome concentrations (125 and 250 µL/mL)? Would this reduction be considered a limitation of the formulation? To enhance this section, provide a more detailed discussion, including potential mechanisms behind the cytotoxicity at higher concentrations. Additionally, strengthen the comparison with existing literature by highlighting similar findings or contrasting trends and their implications for the application of these nanoliposomes.
Ans: Thank you for your insightful feedback regarding Figure 6 and the biocompatibility data. We appreciate the opportunity to provide a more detailed discussion of the observed decrease in cell viability at higher liposome concentrations with references.
“The reduction in cell viability at higher liposome concentrations may be attributed to several potential mechanisms, including phospholipid concentration, oxidative stress, physical crowding, and nutrient deprivation. Elevated concentrations of liposomes result in higher levels of phospholipids, which could disrupt cellular membranes and lead to cytotoxicity. This is consistent with previous studies reporting concentration-dependent cytotoxic effects of lipid-based carriers at higher doses. For instance, lipid nanocapsules have been shown to exhibit toxicity at high concentrations, with their effects varying in a cell-type-dependent manner [41]. Similar trends have been observed in liposomal delivery systems, such as anti-tubercular agent-loaded liposomal vesicles [42], where cytotoxic effects were attributed to the same factors.
Notably, the cytotoxicity threshold often varies depending on cell type, liposome composition, and incubation conditions, highlighting the need for further investigation of our specific formulation [43]. At higher concentrations, the interaction of liposomes with cell membranes could induce oxidative stress, impairing cellular function and viability. This effect is likely influenced by the intricate mechanisms underlying the in-traction between liposomes and cell membranes, which vary depending on the com-position and characteristics of the liposomes [44]. While premium RPO-loaded nanoliposomes contain β-carotene with known antioxidant properties, the empty nanoliposomes lack such bioactives, potentially making them more prone to oxidative damage at elevated doses. In addition to these mechanisms, high liposome concentrations in the culture medium may alter the local environment, leading to physical crowding or reduced nutrient and oxygen availability for the cells, which could impair growth.
Although high concentrations may present challenges, it is important to note that the effective concentrations required for functional applications, such as delivering bioactive compounds, are often below the threshold that induces cytotoxicity. This underscores the potential of RPO-loaded nanoliposomes for safe use when appropriately dosed.”
- - In line 470 it is mentioned that empty nanoliposome displayed no 470 NO inhibition, but some researchers reported that phospholipids may have an added anti-inflammatory action. Did the authors consider their results to not be in accordance with this, or that without loaded bioactive compounds there is no added antiinflamatory activity. Try to incorporate your results with literature more cohesively. This also go for next paragraph, which seems to be a bit nepoveziv.
- - In line 470, it is stated that empty nanoliposomes exhibited no NO inhibition; however, studies that you have mentioned (41) said that phospholipids may have an added anti-inflammatory action. Do the authors interpret their findings as being inconsistent with these reports, or do they attribute the lack of anti-inflammatory activity to the absence of bioactive compounds in the empty nanoliposomes? Consider integrating your results more cohesively with the existing literature, addressing potential reasons for this discrepancy. This recommendation also applies to the following paragraph, which appears somewhat disconnected. Strengthen the narrative flow and ensure clear connections between the results and relevant literature.
Ans: Thank you for your comment and for highlighting the need to better integrate our findings with the existing literature. We acknowledge that some researchers have reported that phospholipids may exhibit anti-inflammatory activity. However, our results indicated that empty nanoliposomes did not display NO inhibition, suggesting that their anti-inflammatory potential might depend on specific conditions or components. We interpret this apparent discrepancy as a result of differences in the context of our study. In our case, the empty nanoliposomes lacked any bioactive compounds, which could explain the absence of observable anti-inflammatory activity. Furthermore, it is possible that the concentration of phospholipids used in our formulation was insufficient to independently elicit an anti-inflammatory response. This aligns with findings in the literature that highlight the dose-dependent nature of phospholipid activity in showing anti-inflammatory action (Jang et al., 2022). The revision was made accordingly and the reference was added.
“All RPO-loaded nanoliposomes demonstrated anti-inflammatory activity by inhibiting NO production (Fig. 7), whereas the empty nanoliposomes showed no such effect. Mokdad et al. [45] reported that phospholipids found in the liposomal membrane may have an added anti-inflammatory action, potentially enhancing their bioavailability. While certain studies have established that phospholipids possess anti-inflammatory properties, the results of this study indicated that empty nanoliposomes did not inhibit NO production. This suggests that their anti-inflammatory potential might be influenced by specific conditions or the presence of additional components. The lack of observable anti-inflammatory activity in empty nanoliposomes can be attributed to the absence of bioactive compounds, which are likely essential for eliciting such effects. Moreover, the concentration of phospholipids in the formulation may have been insufficient to independently trigger an anti-inflammatory response. This finding is consistent with the literature, which highlights the dose-dependent nature of phospholipid activity in exhibiting anti-inflammatory properties [46]. According to Li et al. [47], β-carotene regulates tissue metabolism, which includes microbial flora, oxidative stress, and resistance to inflammation. It interacts with several inflammatory targets, making it a potential therapeutic agent for a variety of diseases. β-carotene can reduce inflammation by inhibiting the production of NO, prostaglandin E2, and superoxide dismutase [47]. It also reduces the expression of iNOS/cox-/NADPH oxidase proteins and mRNA, and inhibits TNF-α levels [48]. β-Carotene can boost the number of NK cells in Wistar rats' blood, increase IL-2 and TNF-α levels, increase GSH-Px in the liver, inhibit tumor cell growth, and lower ALT and AST levels in mice with liver cancer [49]. Li et al. [47] noticed that β-carotene reduced LPS-induced oxidation in porcine intestinal epithelial cells, inhibited Caspase-3 expression, and reduced inflammatory responses in macrophages by inhibiting NF-kB, JK2/STAT3, and JNK2/p38MAPK signaling pathways.”
- - In the conclusions, please include a discussion of the practical applicability of the obtained results, emphasizing their potential impact or real-world use. Additionally, outline directions for future research, such as addressing current limitations, exploring complementary approaches, or testing the findings in broader or more applied contexts.
Ans: Done as requested. “The premium RPO was successfully encapsulated inside a nanoliposome made of egg phosphatidylcholine (EYPC), cholesterol, and Tween 80. The ratio of EYPC to premium RPO was critical for encapsulation. The optimal ratio for EYPC to premium RPO was 7:1, which resulted in the highest encapsulation efficiency and the smallest average particle size. The oxidation stability of RPO was significantly improved after nanoencapsulation with the appropriate EYPC ratio. The RPO-loaded nanoliposomes had a spherical shape with a uniform size distribution. RPO-loaded nanoliposomes with varying EYPC ratios demonstrated anti-inflammatory activity without cytotoxic effects against RAW264.7 cells. Furthermore, the current study showed that RPO-loaded nanoliposomes had longer-term bioactivity than empty nanoliposomes. These findings highlight nanoliposomes as a promising delivery system for enhancing the biological activity and storage stability of premium RPO. The results of this study have practical applicability in the development of functional foods and nutraceuticals, particularly in improving the stability and bioavailability of lipid-soluble bioactives such as RPO. The successful encapsulation of RPO into nanoliposomes suggests potential use in industries where oxidative stability and sustained bioactivity are critical, such as in dietary supplements, food fortification, and pharmaceutical formulations.
Future research should address current limitations by further investigating the stability and bioactivity of RPO-loaded nanoliposomes under diverse environmental conditions, such as heat, light, and pH, as well as during long-term storage. Such studies would provide valuable insights into their robustness and practical applicability. Additionally, exploring complementary approaches, including the incorporation of other bioactive compounds or antioxidants into the nanoliposomes, could enhance their functionality and broaden their potential applications. Testing these nanoliposomes in broader contexts, such as in vivo models or within complex food matrices, will be essential to validate their efficacy, safety, and suitability for real-world use.”

Reviewer 2 Report
Comments and Suggestions for Authors
The article “Design and Bioanalysis of Nanoliposome Loaded with Premium Red Pal Oil for Improved Nutritional Delivery and Stability” states that premium Red Palm Oil (RPO) is an adequate source of vitamins; however, carotenoids are unstable and prone to oxidation due to light or temperature. Therefore, encapsulation of RPO inside nanoliposomes made of egg yolk phosphatidylcholine (EYPC) cholesterol, and Tween 80 is proposed. RPO nanoliposomes were characterized based on their size distribution, polydispersity index, zeta potential, encapsulation efficiency, loading content, nanoliposome morphology, and phospholipid changes. The optimal ratio EYPC/RPO was 7:1. RPO nanoliposomes with varying EYPC ratios had anti-inflammatory activity without showing toxicity in cell lines. The study is interesting and relevant to the food field, I have just suggest to attend these observations
1. Abstract: provide the name for the acronym DPPH•.
2. The introduction section provides an overview of processes to obtain RPO, most harming the biological components. Then liposomes are introduced as a potential alternative to encapsulate and increase the bioavailability of diverse lipophilic and hydrophilic molecules. Finally, the methods used to characterize the RPO/liposomes are listed. However, it does not provide information about the relevance of cell lines, especially the RAW264.7
A) What type of cells are commonly used to test cytotoxicity, and why is important to perform such an assay?
B) Why is relevant the cell line RAW264.7, in this study?
3 Figure 2 y-label, what is transittance? Do you mean transmittance? In the same figure, provide the (%) values in the y-label.
Author Response
Reviewer 2
The article “Design and Bioanalysis of Nanoliposome Loaded with Premium Red Pal Oil for Improved Nutritional Delivery and Stability” states that premium Red Palm Oil (RPO) is an adequate source of vitamins; however, carotenoids are unstable and prone to oxidation due to light or temperature. Therefore, encapsulation of RPO inside nanoliposomes made of egg yolk phosphatidylcholine (EYPC) cholesterol, and Tween 80 is proposed. RPO nanoliposomes were characterized based on their size distribution, polydispersity index, zeta potential, encapsulation efficiency, loading content, nanoliposome morphology, and phospholipid changes. The optimal ratio EYPC/RPO was 7:1. RPO nanoliposomes with varying EYPC ratios had anti-inflammatory activity without showing toxicity in cell lines. The study is interesting and relevant to the food field, I have just suggest to attend these observations
- Abstract: provide the name for the acronym DPPH•.
Ans: Done as requested.
- The introduction section provides an overview of processes to obtain RPO, most harming the biological components. Then liposomes are introduced as a potential alternative to encapsulate and increase the bioavailability of diverse lipophilic and hydrophilic molecules. Finally, the methods used to characterize the RPO/liposomes are listed. However, it does not provide information about the relevance of cell lines, especially the RAW264.7
Ans: Detailed information regarding the cytotoxicity assay is provided in the Methods section.
“2.4.1. Cytotoxicity Assay
RAW264.7 cells are a well-known murine macrophage cell line that is frequently used to assess the possible toxicity of different substances on immune cells. They are an appropriate model for examining the effects of liposome formulations on cellular viability and function because of their capacity to replicate essential macrophage functions, including phagocytosis and cytokine generation [22, 23].”
- A) What type of cellare commonly used to test cytotoxicity, and why is important to perform such an assay?
Ans: Since the MTT test is used to measure cellular metabolic activity as an indicator of cell viability, proliferation, and cytotoxicity, it was used in the present study. For this reason, in vitro screens using the RAW264.7 murine macrophage cell line are widely performed. Therefore, as stated in the manuscript, we exclusively used RAW264.7 cells in our investigation.
- B) Why is relevant the cell line RAW264.7, in this study?
Ans: We selected the RAW264.7 cell line for our cytotoxicity studies due to its relevance as a well-characterized murine macrophage model. These cells exhibit key macrophage functions, making them suitable for investigating immune responses and the effects of various substances on these critical cells. Additionally, their ease of culture and extensive use in prior research facilitate data interpretation and comparison with existing findings.
3 Figure 2 y-label, what is transittance? Do you mean transmittance? In the same figure, provide the (%) values in the y-label.
Ans: Done as request. However, because there were merged spectra from multiple treatments, the percentage value was not provided.

Reviewer 3 Report
Comments and Suggestions for Authors
Dear Autors
The manuscript is an interesting scientific contribution to the study of nanoliposomes as a delivery system for biologically active ingredients and their stability protection. The paper has high scientific level, the experiment is well designed, the discussion is consistent and the final conclusions are interesting. Therefore, the manuscript may be published in FOODS after minor revision.
Suggestions for edition as well as some comments are the following:
Keywords
Please change these keywords “red palm oil; nanoliposome” because they are presented in the title.
Material and methods
LINE 103 Change the units "ppm" to "mg/kg" will be analogous to the remaining tocopherols and β-carotene
Figures 6 and 7
The resolution of the graphs is too low, e.g., the characters of the hashtag ### are blurred. In addition, removing the frames will improve readability
Author Response
Reviewer 3
Dear Autors
The manuscript is an interesting scientific contribution to the study of nanoliposomes as a delivery system for biologically active ingredients and their stability protection. The paper has high scientific level, the experiment is well designed, the discussion is consistent and the final conclusions are interesting. Therefore, the manuscript may be published in FOODS after minor revision.
Suggestions for edition as well as some comments are the following:
Keywords
Please change these keywords “red palm oil; nanoliposome” because they are presented in the title.
Ans: Done as requested.
Material and methods
LINE 103 Change the units "ppm" to "mg/kg" will be analogous to the remaining tocopherols and β-carotene
Ans: Done as requested.
Figures 6 and 7
The resolution of the graphs is too low, e.g., the characters of the hashtag ### are blurred. In addition, removing the frames will improve readability
Ans: Thank you very much. We will retain the current frame settings to maintain clarity and separation between figures. Regarding resolution, the current images meet the requirements for online publication as they can be enlarged without significant loss of quality.

Round 2
Reviewer 1 Report
Comments and Suggestions for Authors
Dear authors,
I am satisfied with the revisions made in response to the provided feedback. The comments and suggestions have been thoroughly and effectively addressed, demonstrating a clear effort to improve the quality of the manuscript. In its current form, I believe the manuscript meets the necessary standards and is well-prepared for publication.